# Identification of a monoclonal antibody that targets PD-1 in a manner requiring PD-1 Asn58 glycosylation

Mingzhu Wang[1,2,5], Junchao Wang[1,2,5], Rongjuan Wang[3,5], Shasha Jiao[3], Shuang Wang[3], Jinchao Zhang ⓘ [3,4]* & Min Zhang ⓘ [1]*

Programmed cell death 1 (PD-1) is inhibitory receptor and immune checkpoint protein. Blocking the interaction of PD-1 and its ligands PD-L1/ L2 is able to active T-cell-mediated antitumor response. Monoclonal antibody-based drugs targeting PD-1 pathway have exhibited great promise in cancer therapy. Here we show that MW11-h317, an anti-PD-1 monoclonal antibody, displays high affinity for PD-1 and blocks PD-1 interactions with PD-L1/L2. MW11-h317 can effectively induce T-cell-mediated immune response and inhibit tumor growth in mouse model. Crystal structure of PD-1/MW11-h317 Fab complex reveals that both the loops and glycosylation of PD-1 are involved in recognition and binding, in which Asn58 glycosylation plays a critical role. The unique glycan epitope in PD-1 to MW11-h317 is different from the first two approved clinical PD-1 antibodies, nivolumab and pembrolizumab. These results suggest MW11-h317 as a therapeutic monoclonal antibody of PD-1 glycosylation-targeting which may become efficient alternative for cancer therapy.

[1] School of Life Sciences, Anhui University, 230601 Hefei, Anhui, China. [2] Institutes of Physical Science and Information Technology, Anhui University, 230601 Hefei, Anhui, China. [3] Beijing Kohnoor Science & Technology Co., Ltd., 102206 Beijing, China. [4] Mabwell (Shanghai) Bioscience Co., Ltd., 201210 Shanghai, China. [5] These authors contributed equally: Mingzhu Wang, Junchao Wang, Rongjuan Wang. *email: zjc@bjkohnoor.com; zhmin07@ahu.edu.cn

Programmed cell death protein 1 (PD-1; also known as CD279), a member of the T-cell receptor CD28 family, is expressed on the surfaces of a variety of immune cells, including monocytes, T cells, and B cells[1]. PD-1 is an important immune checkpoint molecule, which inhibits the functions of CD4[+] and CD8[+] T cells in the tumor microenvironment[2,3]. The key ligands of PD-1 include PD-L1 (also known as CD274, B7-H1) and PD-L2 (also known as CD273, B7-DC); these ligands are expressed by immune cells and can be induced in many different tissues[4–6]. When PD-L1 or PD-L2 binds to PD-1 expressed on the surface of a T cell, the T cell receives an inhibitory signal, causing it to suppress T-cell proliferation and cytokine production, and thereby inhibiting the T-cell-associated immune response[7,8]. PD-L1 and PD-L2 are highly expressed in a variety of human tumor cells, enabling the tumor cells to escape T-cell immune surveillance[9,10]. Blockage of the interaction between PD-1 and PD-L1/PD-L2 improves T-cell function and prevents tumor cells from evading the immune system[11,12]. Therefore, PD-1 and PD-L1/PD-L2 are important therapeutic targets for cancer treatment[13].

Monoclonal antibodies (MAbs) against PD-1 and PD-L1 have been remarkably successful as treatments for various tumors[14,15]. To date, three MAbs against human PD-1 have been globally marketed; these antibodies block the interaction between PD-1 and PD-L1/PD-L2, and reverse PD-1 pathway-mediated immunosuppression[16–19]. Two of these antibodies, nivolumab and pembrolizumab, were approved by the Food and Drug Administration of the United States in 2014 for the treatment of melanoma, non-small cell lung cancer, and other tumors[16–18]. In 2018, the third PD-1 monoclonal antibody, cemiplimab, was approved in the United States for the treatment of metastatic cutaneous squamous cell carcinoma and locally advanced cutaneous squamous cell carcinoma[19].

The co-crystal structures of PD-1/PD-L1 or PD-1/PD-L2 have provided important information and structural basis of molecular interactions between these molecules[20–22]. The crystal structures of PD-1/pembrolizumab complex[23–25] and PD-1/nivolumab complex[24,26] were also reported. These studies provided the basis for the discovery of the precise epitopes and binding modes of antigen–antibody interactions, which, in turn, revealed their antitumor mechanisms. Both antibodies, nivolumab and pembrolizumab, belong to the IgG4 subclass, and their binding depends on the interactions with the flexible loops of PD-1, and the epitopes include residues in the PD-L1-binding site. Although there is some overlap between the two antibodies, both effectively block the interaction of PD-1 with PD-L1.

PD-1 is a glycoprotein that contains four putative N-linked glycosylation sites in its extracellular IgV domain. Studies evaluating PD-1 fucosylation revealed that the four sites of PD-1 (N47, N58, N74, and N116) are all glycosylated, and glycosylation is linked with the functional localization of PD-1[27]. Previously reported PD-1 structures, produced from *Escherichia coli*, provided little data on the glycosylation pattern[20–25]. In early researches, the PD-1/nivolumab complex structure is the only one where the N-glycosylation of PD-1 was observed. In that study, authors demonstrated that N-glycosylation of PD-1 did not affect the binding affinity to nivolumab[26]. Recently, there were two reports about MAbs targeting PD-1, in which the binding modes were also glycosylation-independent[28,29].

The epitope recognized by an antibody is an important characteristic of that antibody. Although existing antibodies block interactions between PD-1 and PD-L1 (or PD-L2), the antigenic epitopes targeted by these antibodies differ[26,30]. Studies focused on discovering specific binding sites or identifying interactions between MAbs and their target antigens may improve the binding specificity and efficiency of the MAbs. Here, we used hybridoma screening and antibody humanization to identify a high-affinity antibody, MW11-h317, which specifically binds human PD-1 and effectively blocks PD-1/PD-L1 and PD-1/PD-L2 interactions. In addition, MW11-h317 has a unique N-linked glycosylation antigen-binding site with antitumor activity. Our study not only helps to provide a deeper understanding of the mechanism behind the human PD-1/antibody interaction but also highlights a potential therapeutic strategy to improve the efficacy of the checkpoint blockade treatment by targeting PD-1 glycosylation.

## Results

### Obtaining recombinant monoclonal antibody MW11-h317.

We immunized mice with the recombinant human PD-1 protein (residues 21-167) and identified the murine monoclonal antibody m317 that bound to human PD-1 protein and inhibited interactions of PD-1 and ligands from hybridoma. We then obtained humanized antibody MW11-h317 according to humanization of m317 antibody frameworks sequences, and without altering affinity and specificity than m317. Finally, we cloned the MW11-h317 antibody sequence into a eukaryotic expression vector, and inserted this vector into Chinese hamster ovary (CHO) cells via electroporation to obtain a cell line stably expressing the humanized antibody.

### Overall structure of PD-1 in complex with MW11-h317 Fab.

The complex structure of human PD-1 extracellular domain (residues 21-167) with MW11-h317 Fab fragment was determined and refined to a resolution of 2.9 Å, with $R_{work}$ and $R_{free}$ values of 0.207 and 0.247, respectively. PD-1 and MW11-h317 Fab formed a 1:1 complex. Human PD-1 assumes a canonical β-sandwich immunoglobulin variable (IgV) topology with a disulfide bond between Cys54 and Cys123 residues. The MW11-h317 Fab in the complex was typical complementarity-determining region (CDR) structure parameters (Fig. 1).

The interaction of PD-1 with MW11-h317 Fab buried ~2370 Å$^2$ of the surface area (the average value of two complexes in the asymmetric unit). The binding interface was formed by residues from the loops of the IgV domain of hPD-1, including the BC loop, C′D loop, and FG loop. Importantly, all of the six CDRs of VH and VL of MW11-h317 Fab contributed to interaction with PD-1. We observed that the residue Asn58 was glycosylated with

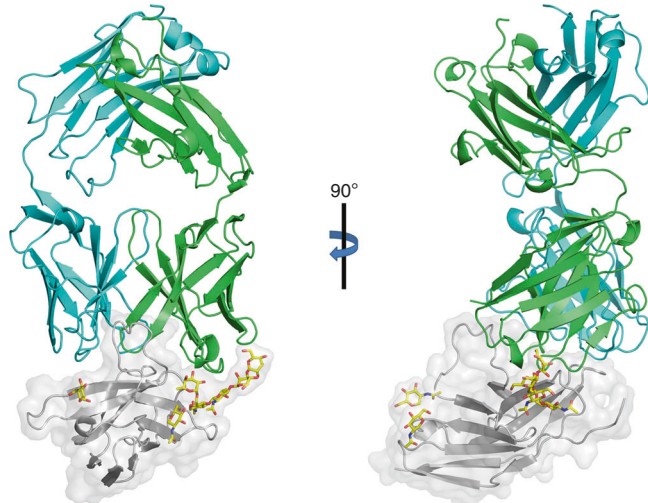

**Fig. 1** The complex structure of MW11-h317 Fab bound to PD-1. The Fab fragment of MW11-h317 and PD-1 are shown as cartoon (MW11-h317 Fab heavy chain in green, light chain in cyan, PD-1 in gray). The glycans in PD-1 are shown as sticks in yellow

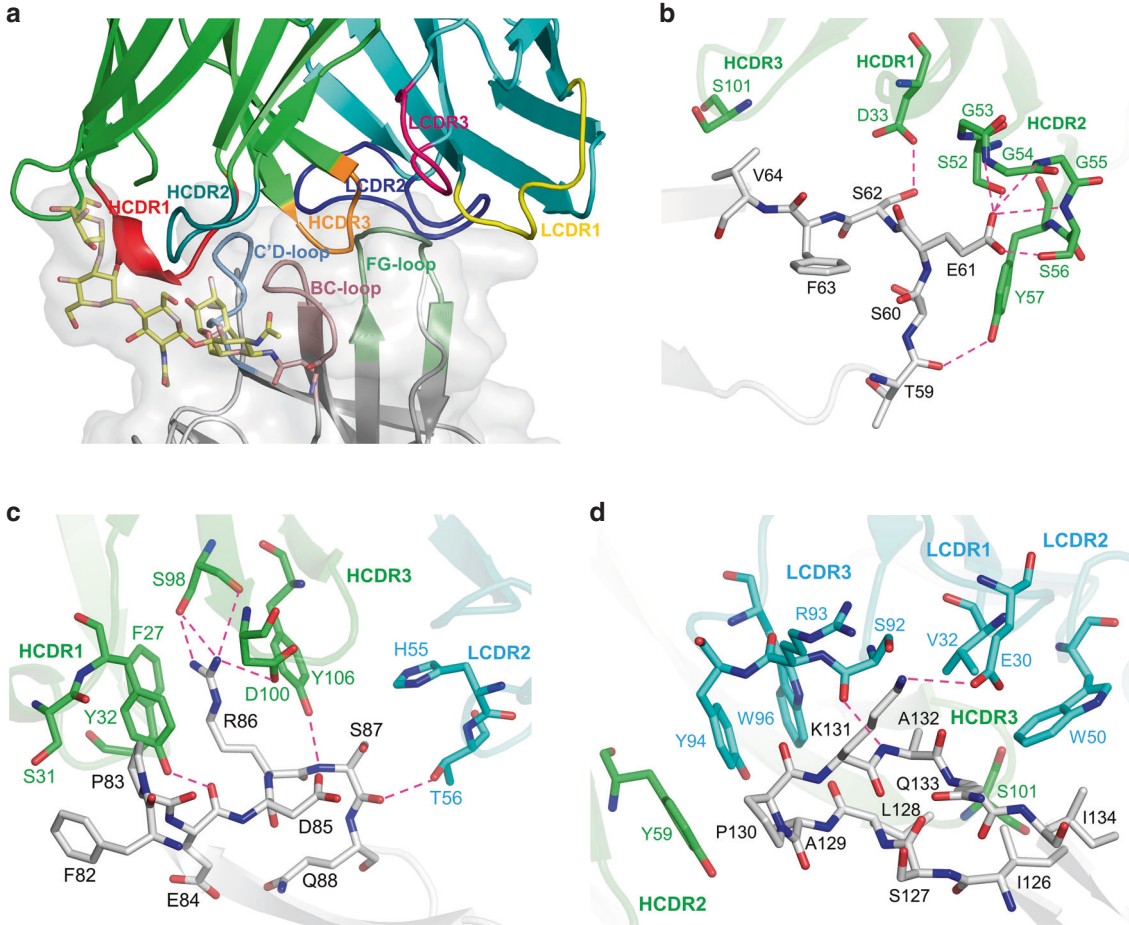

**Fig. 2** Loops of PD-1 in MW11-h317 Fab recognition. **a** The C′D loop, BC loop, and FG loop of PD-1 are represented in skyblue, pink, and lime, respectively. The CDR1, CDR2, and CDR3 loops of MW11-h317 light chain (LCDR1, LCDR2, LCDR3) are shown in yellow, blue, and magenta, respectively. The CDR1, CDR2, and CDR3 loops of MW11-h317 heavy chain (HCDR1, HCDR2, HCDR3) are represented in red, deep teal, and orange, respectively. Detailed interactions of MW11-h317 Fab binding to the PD-1 BC loop **b**, C′D loop **c**, and FG loop **d** are presented. Residues involved in the hydrogen bond interaction are shown as sticks and labeled. Hydrogen bonds are shown as dashed red lines

two N-acetylglucosamines, one fucose and two mannoses, and was present near the interface of PD-1/ MW11-h317 Fab complex, which was involved in the interaction with MW11-h317 Fab (Fig. 1).

**Loops of PD-1 in MW11-h317 Fab recognition.** Structural analysis revealed that the loops of PD-1 contributed to the interaction with MW11-h317 Fab (Fig. 2a). The BC loop residues E61 and S62 formed hydrogen bonds with HCDR1 (D33) and HCDR2 (S52, G53, G54, G55, and S56) residues, while the V64 of BC loop formed Van der Waals interactions with HCDR2 (S101) (Fig. 2b). Specifically, the side chain of R86 in the C′D loop of PD-1 entered the hydrophobic pocket formed by F27 and Y32 in HCDR1, Y106 in HCDR3, and V2 of heavy chain. The guanidino group of R86 stacked with Y32 and formed five hydrogen bonds with S98 and D100 in HCDR3. Moreover, we observed that one ω-guanidino nitrogen atom interacted with the aromatic ring of F27 (Fig. 2c). The FG loop, mainly consisting of hydrophobic residues including I126, L128, A129, P130, A132, and I134, interacted with a hydrophobic pocket formed by V32 in LCDR1, Y91, Y94, W96 in LCDR3, W50 of light chain, and Y59 of heavy chain (Fig. 2d).

**An intact Asn58 glycosylation site is crucial for MW11-h317 binding to PD-1.** Human PD-1 is a transmembrane glycoprotein

which belongs to the Ig superfamily. There are four predicted N-linked glycosylation sites (N49, N58, N74, and N116) in the IgV domain (Fig. 3a). In the PD-1/MW11-h317 Fab complex structure elucidated in our experiment, N49, N58, and N116 were glycosylated, which was determined from the clear electron density. However, the sugar chain of N74 was ambiguous. The N58 site of PD-1, the only glycosylation site near the interface of PD-1/MW11-h317 Fab complex, was glycosylated with two N-acetylglucosamines, one fucose and two mannoses (Fig. 3a), while only one N-acetylglucosamine could be modeled from the electron density of N49 and N116. The glycans linked to N58 are a common "complex" type N-glycan structures observed in mammalian cells[31] (Fig. 3b, c). The glycans linked to N58 made direct interactions with V domain of the heavy chain, burying ~500 Å² of surface area (the average value of two complexes). The acetyl group of the first N-acetylglucosamine formed a hydrogen bond with a hydroxyl group of Y57 of the heavy chain. The second N-acetylglucosamine interacted with G54 of the heavy chain through the Van de Waals interactions. The 2-hydroxyl group and 3-hydroxyl group of the fucose formed hydrogen bonds with the amino group of G54 and the carbonyl group of S31, respectively. The 2-hydroxyl group of the first mannose formed a hydrogen bond with the carbonyl group of G54, while the 6-hydroxyl group of the second mannose formed a hydrogen bond with the carbonyl group of R72 (Fig. 3d). Comparing the glycosylated hPD-1 we elucidated in this complex to the protein that

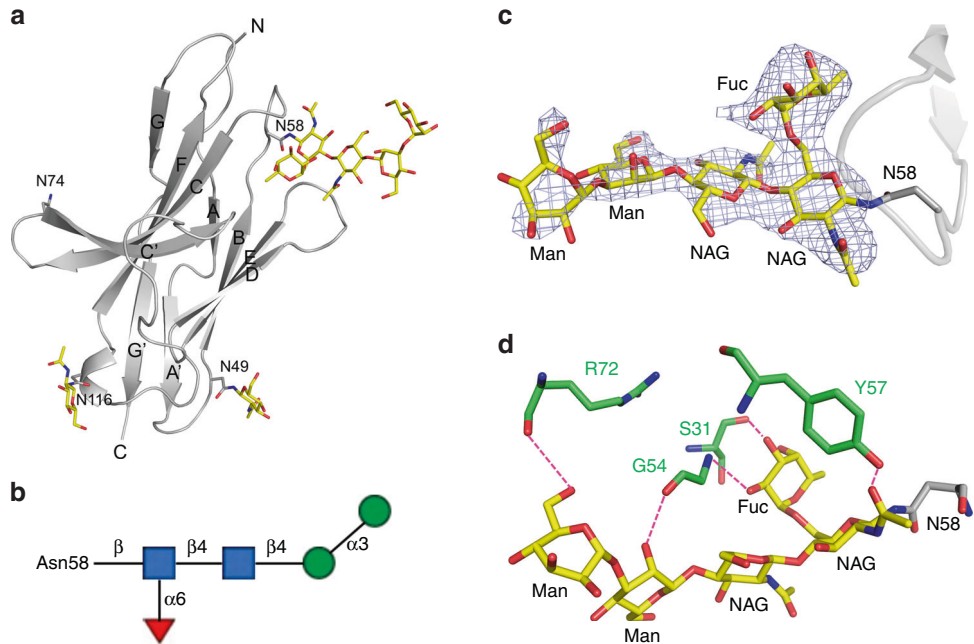

**Fig. 3** Glycosylation of PD-1. **a** N-glycosylation of N49, N58, and N116 in PD-1. **b** Diagrammatic representation of N-link glycans of N58, where blue squares represent N-acetylglucosamines, red triangle is fucose, and green circles are mannose. **c** The Fo−Fc simulated annealing omit map of N-linked glycans of N58 contoured at $3.0\sigma$. The stick model of glycans was superposed in the image. **d** Interactions between N-linked glycans of N58 and MW11-h317. The participating amino acid residues and glycans are shown with a stick model, with amino acids in MW11-h317 colored in green and glycans in PD-1 colored in yellow. Hydrogen bonds are indicated with magenta dashed lines

was previously reported and lacked glycan, we determined that the glycosylation had no effect on the overall structure or conformational change of PD-1. Therefore, PD-1 glycosylation of N58 is likely involved in the MW11-h317 binding by direct interaction, and not via induction of conformational changes.

Next, we further investigated the effects of N-glycosylation of PD-1 on MW11-h317 binding using the alanine scanning mutants, with nivolumab and pembrolizumab as controls. We expressed WT or mutant PD-1 proteins in HEK 293 cells, incubated the samples with the anti-PD1 MAbs and analyzed them by FACS. The results revealed that N58A-mutated PD-1 exhibited significant loss of MW11-h317 binding, while the other three mutants (N49A, N74, and N116) displayed no changes compared with WT protein (Fig. 4). However, all four N-glycosylation sites in mutated PD-1 proteins were capable of binding to nivolumab (Supplementary Fig. 1), in agreement with a previous report[26]. Similarly, binding to pembrolizumab was comparable, where all four mutants had no effect on the interaction with PD-1 (Supplementary Fig. 1). Although we cannot exclude that the mutation of glycolysation sites have effects other than blocking glycosylation, our data strongly suggest that MW11-h317 is a PD-1 glycosylation-dependent antibody, which is entirely different from previously reported therapeutic antibodies.

**MW11-h317 competitive binding with PD-L1 or PD-L2.** Structural superposition of PD-1/MW11-h317 Fab complex and PD-1/PD-L1 complex (PDB ID: 4ZQK) revealed that MW11-h317 Fab and PD-L1 interacted with PD-1 through overlapping surface regions (Fig. 5), suggesting that MW11-h317 and PD-L1 can displace one another from PD-1. The overlapping regions of PD-1 were predominantly located on the FG loop. The FG loop residues of PD-1 (I126, L128, A132, and I134) and V64 in the BC loop created a hydrophobic region with the residues of PD-L1 (I54, Y56, M115, A121, and Y123), and the hydrophobic amino

acids in the BC and FG loops also interacted with the hydrophobic residues in VH (Y59) and VL (V32, W50, Y91, Y94, and Y96) of MW11-h317 Fab. These results indicate that the blocking mechanism of MW11-h317 is dependent on the CDRs of MW11-h317 Fab, which introduces a steric clash to abolish the binding of PD-L1 to hPD-1. Superposition of PD-1/MW11-h317 Fab complex with PD-1/PD-L2 complex (PDB ID: 3BP5) revealed that MW11-h317 could block the PD-L2 to PD-1 in the same manner (Supplementary Fig. 2).

**Distinct epitopes compared with pembrolizumab and nivolumab.** To date, five PD-1/Mab complex structures have been reported, including PD-1/nivolumab Fab[24,26], PD-1/pembrolizumab Fab[23–25], and three other PD-1/Fabs[28,29]. Previous studies have shown that each PD-1 Mab exhibits different binding modality to PD-1. To compare the difference mechanisms of MAbs binding to and blocking of PD-1, we compared our PD-1/MW11-h317 Fab complex structure with PD-1/nivolumab complex structure (PDB ID: 5WT9) and PD-1/pembrolizumab complex structure (PDB ID: 5GGS), as well as the PD-1/PD-L1 complex structure (PDB ID: 4ZQK).

Collectively, we discovered that the three MAbs bind to PD-1 from different orientations. The results revealed a stereo clash with PD-L1, with the different epitopes contributing from the VH and VL among the MAbs (Supplementary Table 1). The blocking mechanism of nivolumab targeting the interaction of PD-1/PD-L1 was mainly contributed by the VH domain, while both the VH and VL of pembrolizumab and MW11-h317 were involved in the blocking of PD-1/PD-L1 interaction. The recognition sites of the three MAbs to PD-1 are different. Specifically, nivolumab interacts with the N-loop of PD-1, while pembrolizumab interacts with the C′D loop of PD-1. The binding of PD-1 with MW11-h317 was localized to the C′D loop, and the N58 glycosylation on the BC loop also contributed to the interaction with PD-1. The MW11-h317 showed an overlap in PD-1-binding areas compared

**Fig. 4 N-glycosylation of N58 participates in MW11-h317 recognition.** A flow cytometric analysis of MW11-h317 binding to WT PD-1 or various glycosylation sites mutated proteins (N49A, N58A, N74A, and N116A) expressed on the cell surface of HEK 293 cells. Plasmids expressing WT PD-1 or mutant proteins fused with EGFP were used for transfection. Mock-transfected HEK 293 cells were used as negative control (NC)

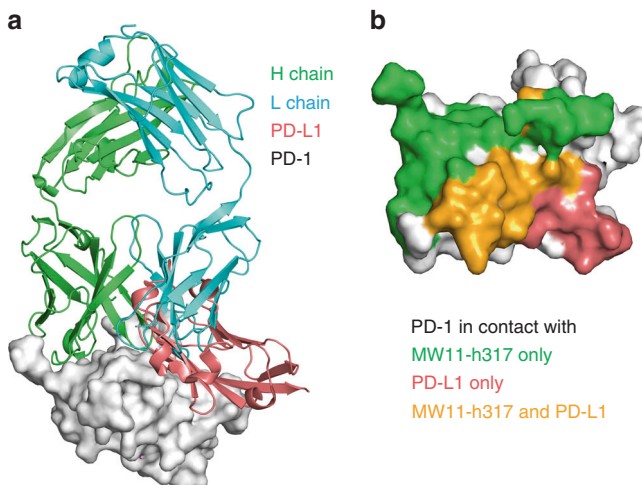

**Fig. 5 Competitive binding of MW11-h317 Fab with PD-1 ligand PD-L1.** **a** Superposition of the PD-1/MW11-h317 Fab complex structure with PD-1/PD-L1 complex structure (PDB ID: 4ZQK). MW11-h317 H chain is shown in green, L chain in cyan and PD-L1 in pink. PD-1 is shown as surface representation in gray. **b** Binding surface of PD-1 with MW11-h317 or PD-L1. The residues in contact with MW11-h317 are shown in green, whereas residues in contact with PD-L1 are shown in pink, and the overlapping residues bounded by both MW11-h317 and PD-L1 are shown in orange

with nivolumab and pembrolizumab, predominantly in the FG loop (Fig. 6).

**Activities of MW11-h317 are comparable to nivolumab.** MW11-h317 specifically bound to human PD-1 protein with an affinity ($K_D$) of $3.55 \times 10^{-9}$ M; the affinity of the control nivolumab was $3.65 \times 10^{-9}$ M. Under the same conditions, MW11-h317 dissociation [$k_d$ (s$^{-1}$): $8.43 \times 10^{-4}$] was slower than nivolumab dissociation [$k_d$ (s$^{-1}$): $1.90 \times 10^{-3}$], while nivolumab association [$k_a$ (M$^{-1}$ s$^{-1}$): $5.21 \times 10^5$] was faster than MW11-h317 [$k_a$ (M$^{-1}$ s$^{-1}$): $2.38 \times 10^5$], which make MW11-h317 has a similar $K_D$ to nivolumab. Those results suggested that MW11-h317 had a high affinity with human PD-1 protein (Fig. 7a).

MW11-h317 effectively blocked the binding of recombinant human PD-1 to its ligands, PD-L1 and PD-L2. Here, we used ELISAs to assess the competitive inhibitory effects of the MW11-h317 and nivolumab on PD-1/ligand binding. The half-maximal inhibitory concentrations (IC$_{50}$) of MW11-h317 and nivolumab for PD-1/PD-L1 binding were 1.4 and 1.3 nM, respectively (Fig. 7b). These results indicated that the inhibitory effects of MW11-h317 on PD-1/ligand binding were similar to those of nivolumab.

Our in vitro experiments by stimulating human mixed lymphocytes indicated that the MW11-h317 increased the secretion of cytokines IL-2 and IFN-γ, the indicators of T cell activation (Fig. 7c), which have been considered as classical features of Th1-type cell-mediated immune response. This

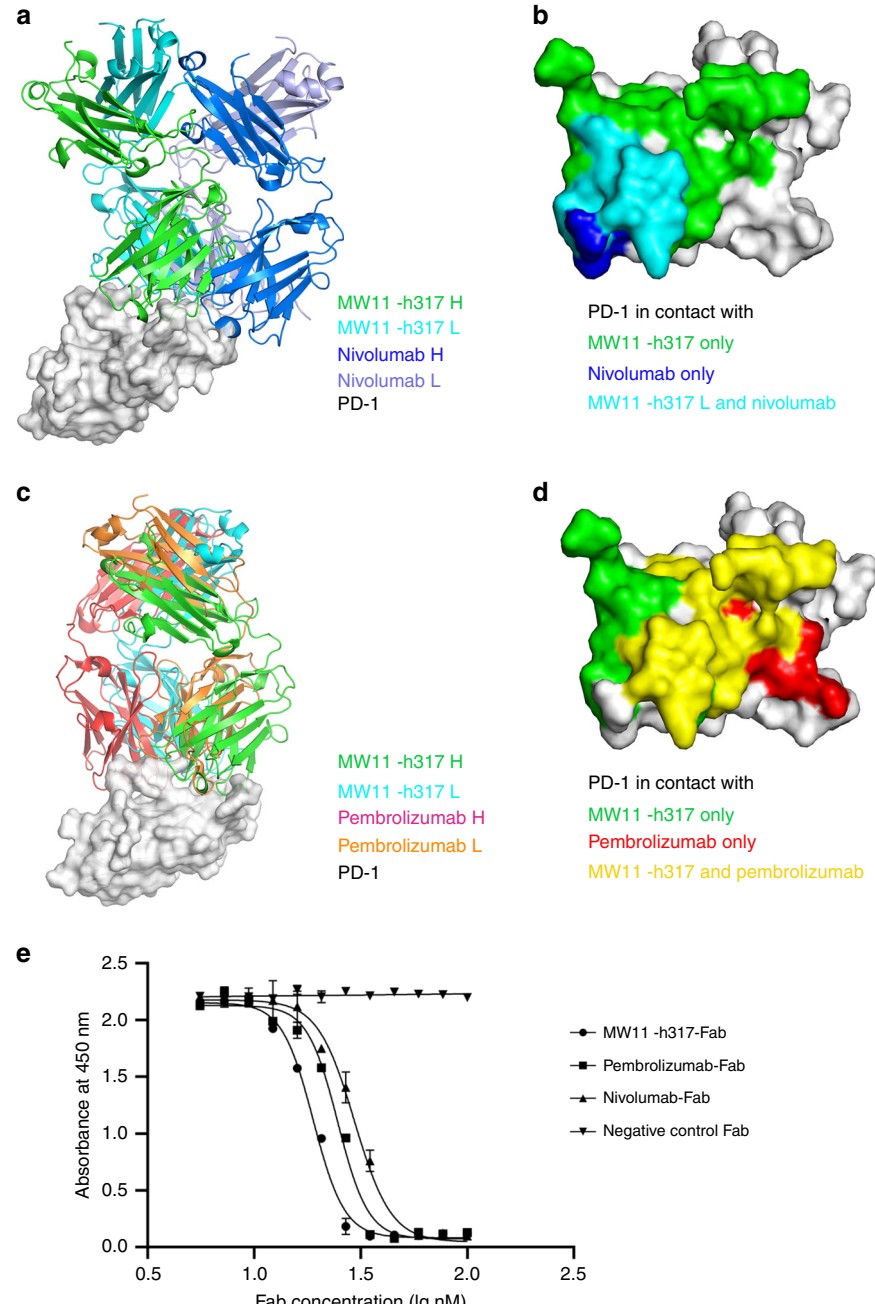

**Fig. 6** Distinct blocking mechanism of MW11-h317 compared with nivolumab and pembrolizumab. **a**, **b** Competitive binding of MW11-h317 and nivolumab (PDB ID: 5WT9) with PD-1. MW11-h317 and nivolumab are shown in green and blue, respectively, and the overlapping residues bounded by both MW11-h317 and nivolumab are shown in cyan. **c**, **d** Competitive binding of MW11-h317 and pembrolizumab (PDB ID: 5GGS) with PD-1. MW11-h317 and pembrolizumab are shown in green and red, respectively, and the overlapping residues bounded by both MW11-h317 and pembrolizumab are shown in yellow. **e** Antibodies competitive ELISA indicated that MW11-h317 was partly competitive with nivolumab or pembrolizumab. $n = 3$ independent experiments; data are mean ± SEM

suggested that MW11-h317 might induce a Th1-type cell-mediated immune response.

We proceeded to test the antibodies in a mouse xenograft model. We used HuGEMMPD-1 mice (with a C57BL/6J genetic background) in this experiment. We inserted the human PD-1 protein-coding region into the ATG position of the mouse model, in order to express human PD-1 instead of mouse PD-1. MC38 is a murine intestinal cancer cell line derived from C57BL/6 mice. Using genetic engineering, mouse PD-L1 was knocked out of MC38 cells. Human PD-L1 was then inserted, generating an MC38 cell line expressing human PD-L1. This cell line was designated MC38-hPD-L1. Our in vivo antitumor experiment demonstrated that both a high dose

(10 mg kg$^{-1}$) and a moderate dose (2 mg kg$^{-1}$) of the MW11-h317 significantly inhibited the growth of subcutaneous xenografts in PD-1 transgenic mice. However, low doses of the MW11-h317 did not have an obvious antitumor effect, suggesting that high and moderate doses of MW11-h317 had antitumor effects comparable to nivolumab (Fig. 7d).

## Discussion

Protein glycosylation as post-translation modification is involved in protein location, stability, and protein–protein interactions[32]. Immune checkpoint proteins are predominantly glycosylated.

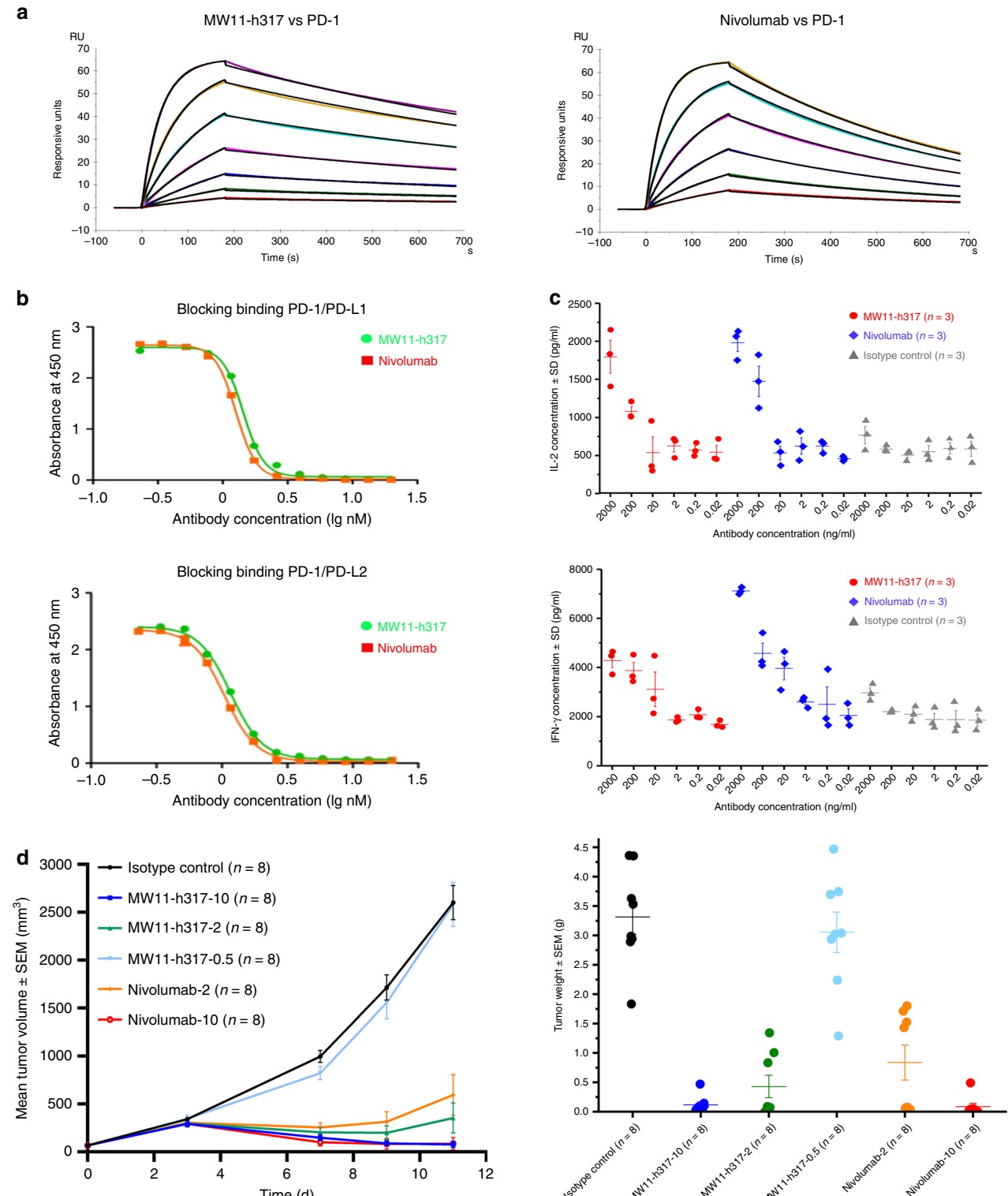

**Fig. 7** Comparable activities of MW11-h317 to nivolumab. **a** Affinities of MW11-h317 and nivolumab to PD-1. **b** MW11-h317 and nivolumab blocked the binding between PD-1 and PD-L1 (or PD-L2). **c** IL-2 or IFN-γ production in mixed lymphocytes stimulated by MW11-h317 and nivolumab. **d** Volumes and weights of subcutaneous xenograft tumors in PD-1 transgenic mice. Administration dose 10, 2, and 0.5 mg kg$^{-1}$ indicated as 10, 2, and 0.5 following antibodies' name ($n = 8$ mice/tumor experiments)

Several studies have focused on exploring the effects of glycosylation on the interactions between the antibody and its target, including tumor-specific glycosylation of MUC1, which is important in antigen recognition by MUC1-specific antibodies. Specifically, the SM3 antibody directly binds to the MUC1 glycan epitope, while the high affinity of AR20.5 antibody to MUC1 was due to the glycosylation, which altered the conformational equilibrium of the antigen[33,34]. In a recent study evaluating triple-negative breast cancer, researchers found that glycosylation was required for the interaction between PD-1 and PD-L1. They

discovered that the antibody targeting glycosylated PD-L1 blocked the PD-L1/PD-1 interaction and promoted PD-L1 internalization and degradation[35].

PD-1 is a transmembrane glycoprotein with four N-linked glycosylation sites (N49, N58, N74, and N116). A previous study elucidating the PD-1/nivolumab Fab complex structure revealed that the binding of nivolumab to PD-1 is not glycosylation dependent[26]. In this study, our PD-1/MW11-h317 Fab complex structure suggested that there were more glycosylated sites observed compared to the PD-1/nivolumab Fab complex structure. Moreover, structural comparison suggested that glycosylation had no influence on the PD-1 conformation but was involved in the interaction with MW11-h317. Overall, we discovered that N58 sugar motif is the component of PD-1 epitope targeted by MW11-h317, which is significantly different from interactions of PD-1 with pembrolizumab and nivolumab.

PD-1 is one of the pivotal molecules of the immune checkpoint therapy. Antibodies targeting PD-1 are the most primary molecules for clinical antibodies combination or bispecific antibodies development in the future. Although nivolumab and pembrolizumab have been approved for cancer therapy by FDA, screening of new therapeutic mAbs targeting PD-1 is beneficial for multiple tumors treatments. Structural analysis aids in elucidating the mechanisms of recognition, as well as helps in developing novel antibodies. In our study, MW11-h317 achieves its striking PD-1 specificity by interactions with both the loops and the glycan portions of PD-1, which will likely provide major insights into glycan–antibody interactions that will help guide therapeutic applications.

## Methods

**Animal experiments.** All mice used for the present study were C57BL/6J mice of female (6–8 weeks old). Animal experiments were approved by Institutional Animal Care and Use Committee (IACUC) of Crown Bioscience (Taicang) Inc. The approval number is AN-1803-13-502.

**Expression of human PD-1 for immunization and hybridoma generation.** The human PD-1 protein (residues 21–167) was expressed using the mammalian cell expression system. An N-terminal signal peptide was added to target the protein to the secretory pathway, and a mouse Fc tag was added to facilitate downstream purification. The synthesized, codon-optimized cDNA was cloned into pKN293E vector (constructed in our laboratory) and the protein was expressed in human embryonic kidney 293 (HEK 293) cells using 293fectin reagent (Cat.: 12347019, Life Technologies). Cells were grown in FreeStyle™ 293 Expression Medium (Cat.:12338026, Life Technologies) for 5 days following transfection. The supernatants were concentrated and the protein was purified by Mabselect chromatography column (Cat.: 29-0491-04, GE Healthcare). The purified protein was buffer exchanged into PBS using Sartorius Stedim Vivaspin 500 Centrifugal Concentrators (Cat.: VS0122, Sartorius Stedim Lab) and filtered prior to injection. Protein purity and size were confirmed by SDS–PAGE.

**Obtaining monoclonal antibody MW11-h317.** The candidate murine antibody (m317) was obtained via hybridoma screening and activity analysis. The sequences encoding the variable regions of the light and heavy chains of the murine antibody were cloned to the upstream of the sequences encoding the constant regions of the light and heavy chains of human antibodies (hKappa, hIgG4). The fused antibody was expressed in mammalian cells (HEK 293) using 293fectin reagent (Cat.: 12347019, Life Technologies) to prepare the chimeric antibody. Cells were grown in FreeStyle™ 293 Expression Medium (Cat.:12338026, Life Technologies) for 5 days following infection. The supernatants were concentrated and the protein was purified by Mabselect chromatography column (Cat.: 29-0491-04, GE Healthcare). The purified protein was buffer exchanged into PBS using Sartorius Stedim Vivaspin 500 Centrifugal Concentrators (Cat.: VS0122, Sartorius Stedim Lab) and filtered prior to injection. Protein purity and was confirmed by SDS–PAGE. The lead antibody was then determined. To humanize the antibody sequence, we first selected a humanized template from the Germline database. We tested the activity and function of the humanized antibodies with in vitro cytological experiments. We then inhibited tumor growth in vitro and in vivo to obtain the sequence of humanized PD-1 antibody (MW11-h317), without altering affinity and specificity.

**Expression of recombinant MW11-h317 antibody.** The synthesized, codon-optimized genes for the heavy and light chains of the MW11-h317 were ligated into a single eukaryotic expression vector (constructed in our laboratory). And the recombinant protein (MW11-h317 antibody) was expressed in CHO-K1 cells (ATCC, suspension) as stable lines using the Nucleorfector™ 2b device (VCA-1003, Lonza). Selection was performed with L-methionine sulfoximine (Cat.: M5379, sigma) at concentrations up to 50 μM. Expression was scaled up in CD-CHO (Cat.: 12490025, Gibco) medium. Supernatant containing the antibody was collected after 7–9 days and purified over by Mabselect chromatography column (Cat.: 29-0491-04, GE Healthcare). The purified protein was buffer exchanged into PBS using Sartorius Stedim Vivaspin 500 Centrifugal Concentrators (Cat.: VS0122, Sartorius Stedim Lab). Protein purity and size were confirmed by SDS–PAGE and SEC-HPLC.

**Protein purification and complex preparation.** The gene encoding the extracellular domain of PD-1 (residues 21–167) was cloned into pKN293E vector (constructed in our laboratory). The plasmid was transfected into CHO cells by electrotransfection (VCA-1003, Lonza) to express the protein. Cells were scaled up and grown in CD-CHO (Cat.: 12490025, Gibco) medium for 15 days following infection. The supernatants were concentrated and the protein was purified by a nivolumab-coupled resin (coupled in our laboratory). The purified protein was buffer exchanged into PBS using Sartorius Stedim Vivaspin 500 Centrifugal Concentrators (Cat.: VS0122, Sartorius Stedim Lab). Protein purity and size were confirmed by SDS-PAGE and SEC-HPLC.

The expression of MW11-h317 Fab was achieved by a co-transfection of two plasmids, encoding variable region of light chain (VL) and variable region of heavy chain (VH) into HEK 293 cells using 293fectin reagent (Cat.: 12347019, Life Technologies). Cells were cultured in FreeStyle™ 293 Expression Medium (Cat.:12338026, Life Technologies) for 5 days following transfection and were then collected. The culture supernatants were purified using Protein G affinity columns (Cat.:17-0404, GE) and then further processed using a HiTrap Capto S (Cat.:17371751, GE Healthcare) to enhance purity. Protein purity and size were confirmed by SDS–PAGE and SEC-HPLC. The PD-1 protein (residues 21–167) and MW11-h317Fab were mixed at a molar ratio of 1:2. The mixed sample was incubated for 30 min on ice and then purified using a HiLoad 16/60 Superdex 200 pg column (Cat.: 28989335, GE Healthcare). Fractions corresponding to the complex were pooled and concentrated to either 8 or 12 mg/ml for crystallization.

**Crystallization, data collection and structure determination.** The PD-1 and MW11-h317 Fab complex were used for crystal screening by vapor-diffusion sitting-drop method carried out at 16 °C. Diffracting crystals were grown in a buffer composed of 0.1 M Bis–Tris (pH 6.5) and 28% polyethylene glycol monomethyl ether 2000. The crystals were flash-frozen in liquid nitrogen with a cryoprotectant with 10% (v/v) ethylene glycol added to the reservoir solution. Diffraction data were collected at Beamline BL17U of the Shanghai Synchrotron Radiation Facility (SSRF) at a wavelength of 0.9792 Å with an ADSC Q315r detector[36]. The data were then processed using HKL2000 software[37]. The structure was resolved by molecular replacement method using PHASER[38] software in CCP4 suite[39], using PD-1 structure (PDB ID: 3RRQ) and nivolumab Fab structure (PDB ID: 5WT9) as the search models. There were two PD-1/317Fab complexes per asymmetry unit. The final model was rebuilt with COOT[40], and refined with REFMAC[41] and PHE-NIX[42]. The $R_{work}$ and $R_{free}$ of the final model are 0.207 and 0.247, respectively. 96.4% of residues are in the most favored region of Ramachandran plots, and 3.6% of residues are in the allowed region. The final model is of good quality with an overall MolProbity score of 1.44. Detailed statistics for data collection and structure determination are summarized in Table 1.

**Flow cytometric analysis of MW11-h317 binding to PD-1 mutants.** Firstly, PD-1 (lacking the intracellular region) was fused with Enhanced green fluorescent protein (EGFP) and then cloned into the **pKN009** vector (constructed in our laboratory). The plasmids expressing PD-1 mutants N49A, N58A, N74A, or N116A were created using site-directed mutagenesis. The plasmids were then transfected into HEK 293 cells using 293fectin reagent (Cat.: 12347019, Life Technologies), and the cells were cultured for 24 h, collected, and resuspended in phosphate buffered saline (PBS) at $1 \times 10^7$ cells ml$^{-1}$. Next, the HEK 293 cells expressing wild-type (WT) PD-1 or PD-1 mutants were stained with anti-PD1 MAbs at room temperature for 30 min, washed three times with PBS and then stained with the secondary antibody (Alexa Fluor® 647 anti-human IgG, #109-605-098, Jackson ImmunoResearch Laboratories, West Grove, PA, USA) for another 30 min. Following a washing step, cells were analyzed by flow cytometry with a Beckman Coulter FACS machine. Antibodies nivolumab (Lot: AAW4553, Bristol-Myers Squibb) and pembrolizumab (Lot: 6SNL81506, Merck &Co.) were also analyzed in the same way.

**Antibody binding kinetics.** The affinity of MW11-h317 and nivolumab was determined via SPR on a Biacore S200 system (GE Healthcare) . Human IgG capture antibody in the standard IgG capture antibody kit (Cat.:BR-1008-39, GE Healthcare) was immobilized on a CM5 chip (Cat.:BR-1005-30, GE Healthcare) using standard amino coupling kit (Cat.:BR-1000-50, GE Healthcare). Antibody

**Table 1 Data collection and refinement statistics[a]**

|  | MW11-h317-PD-1 |
|---|---|
| *Data collection* | |
| Space group | P2 |
| Cell dimensions | |
| $a$, $b$, $c$ (Å) | 102.61, 54.22, 126.08 |
| $\alpha$, $\beta$, $\gamma$ (°) | 90, 113.92, 90 |
| Resolution (Å) | 50.00–2.90 (3.00–2.90)[b] |
| $R_{merge}$ | 0.162 (0.986) |
| $I/\sigma I$ | 8.3 (1.5) |
| Completeness (%) | 99.9 (100.0) |
| Redundancy | 4.1 (4.2) |
| Total/unique reflections | 118,432/28,800 |
| *Refinement* | |
| Resolution (Å) | 50.00–2.90 (2.99–2.90) |
| No. of reflections | 28,784 (2736) |
| $R_{work}/R_{free}$ | 0.207 (0.303)/0.247 (0.360) |
| No. of atoms | |
| Protein | 8408 |
| Ligand/ion | 176 |
| Water | 8 |
| B-factors (Å$^2$) | |
| Protein | 55.2 |
| Ligand/ion | 60.0 |
| Water | 42.6 |
| R.m.s. deviations | |
| Bond lengths (Å) | 0.003 |
| Bond angles (°) | 0.639 |

[a]One crystal was used for this structure
[b]Values in parentheses are for highest-resolution shell

was captured at a certain level (200 Ru here) and reacted with recombinant human PD-1 (residues 21–167) at gradient concentrations (60, 30, 15, and 3.75 nM respectively) in fluid HBSEP buffer (PH 7.4) (Cat.:BR-1006-69, GE Healthcare). At the end of each cycle, the captured antibody, along with PD-1, was washed away with regeneration buffer (3 M MgCl$_2$) and the chip was used for the next cycle reaction until the test was completed. Then, the affinity was calculated in a 1:1 (Langmuir) binding fit model by BIAevaluation Software.

**ELISA detection of the MW11-h317-associated inhibition of PD1 and ligands interactions**. ELISA plates were coated with 0.5 μg mL$^{-1}$ recombinant human PD-1 protein (residues 21–167), and incubated at 4 °C overnight, followed by blocking with 5% bovine serum albumin protein at 37 °C for 60 min. Either MW11-h317 or nivolumab antibodies (starting concentration of 3 μg mL$^{-1}$; 1.5-times serially diluted) were added each microplate well, and allowed to react at 37 °C for 120 min. Next, we added 1 μg mL$^{-1}$ PD-L1-mFc (GenBank accession no. NP_054862.1; residues 19–238; Lot: 20180412) to each well and incubated plates at 37 °C for 60 min. Then, horseradish peroxidase (HRP)-anti-mouse Fc secondary antibodies (1:5000 dilution; Catalog no. 115-035-071; Jackson Immuno Research) were added to each well, and allowed to react for 45 min. Finally, tetramethylbenzidine (TMB, Cat.:ME142, GalaxyBio) substrate was added and allowed to react for 15 min to develop color. The reaction was stopped by adding 2 M HCl. We measured the absorbance of each well in the microplate at a wavelength of 450 nm, using a reference wavelength of 630 nm.

To determine whether MW11-h317 inhibited the binding of PD-1 to PD-L2, we added 2 μg mL$^{-1}$ PD-L2-hFc-Biotin (GenBank accession no. NP_079515.2; residues 20–220; Lot: 2016.12.05) to the antibody, and incubated the mixture at 37 °C for 60 min. We then added HRP-streptavidin (1:2000 dilution; Catalog no. S2438; Sigma-Aldrich) secondary antibodies and allowed the mixture to react 30 min at 37 °C. The remaining steps were as described above.

Antibodies competitive ELISA was performed in the same way. Fabs of antibodies, as competitive phase, were treble gradient diluted from 100 to 5.58 nM. MW11-h317, as detective phase, maintained at 0.5 nM. Fabs and MW11-h317 were mixed and then were added and incubated in ELISA plates coating with recombinant human PD-1 protein (residues 21–167). Finally HRP-anti-human Fc secondary antibody was used to detected how much MW11-h317 binding with PD-1 protein on the ELISA plates.

**MW11-h317-assoctiated stimulation of human T cells in vitro**. *solation of CD14 + monocytes and dendritic cells (DCs)*: We used Ficoll-Paque Plus reagent (Catalog no.: 17-1440-03; Lot: 10246684; GE Healthcare) to extract PBMCs, following the manufacturer's instructions. We used CD14 magnetic beads (Catalog no.: 130-050-

201; Lot: 5170814438; Miltenyi) to select positive CD14+ monocytes, following the manufacturer's instructions. We resuspended cells in ImmunoCult DC Differentiation Medium (Catalog no.: 10988; Lot: 17G83035; STEMCELL) at a density of $5 \times 10^5$ cells/mL. We added 5 mL of the cell suspension to a T-25 cell culture flask and incubated the suspension at 37 °C under 5% CO$_2$ for 3 days. The mixture was then centrifuged and incubated again in fresh medium at 37 °C for 2 days. We then added 50 μL of ImmunoCult Dendritic Cell Maturation Supplement (Catalog no.: 10989; Lot: 17B77271; STEMCELL) to the cells, and incubated the cells at 37 °C for 2 days for the mixed lymphocyte reaction (MLR) assay.

*Isolation of CD4$^+$ T cells*: We used Ficoll-Paque Plus reagent to extract PBMCs, following the manufacturer's instructions. We used a CD4$^+$T cell isolation kit (Catalog no.: 130-096-533; Lot: 5171016623; Miltenyi) to isolate the CD4$^+$ T cells from the PBMCs, following the manufacturer's instructions, for the MLR assay.

*MLR assay*: The test antibodies, MW11-h317 and nivolumab, as well as the human NC-IgG4 control (Lot: AB170090), were each prepared in complete medium at a concentration of 8 μg mL$^{-1}$. We then prepared a 10-fold serial dilution (a total of six concentrations). We added 50 μL diluted antibody solution to the appropriate wells of a 96-well plate. The CD4$^+$T cells were diluted to a density of $1 \times 10^6$ cells/mL, and 100 μL of diluted cell suspension ($1 \times 10^5$ cells per well) was added to appropriate wells. We digested the DCs with 2 mM ethylenediaminetetraacetic acid (EDTA) to digest the DCs, followed by centrifugation at 1500 rpm for 5 min. The DC pellet was resuspended at a density of $2 \times 10^5$ cells/mL. We then added 50 μL of DC suspension ($1 \times 10^4$ cells per well) to the appropriate wells for a total volume of 200 μL per well. The ratio of CD4+ T cells to DCs in each well was 10:1. The microplates were incubated at 37 °C for 5 days, followed by centrifugation. Cell supernatants were collected. We used the interferon gamma (IFN-γ) ELISA kit (Catalog no.: 430106, Lot: B247782, Biolegend) and the interleukin (IL)-2 ELISA kit (Catalog no.: DY202, Lot: P155804, R&D) to determine the IFN-γ and IL-2 concentrations, respectively.

**Antitumor effect of MW11-h317 on mouse xenograft model**. We subcutaneously injected the right sides of 65 HuGEMMPD-1 mice (6–8 weeks old; purchased from Shanghai Model Organisms Center, Inc., Shanghai, China) with MC38-hPD-L1 tumor cells ($1 \times 10^6$ cells per mouse; $1 \times 10^6$ cells resuspended in 100 μL PBS). Once the average tumor volume in the tumor-bearing mice was ~60–100 mm$^3$, we measured the bodyweight of each mouse with a digital scale and the volume of each tumor with vernier calipers. All mice were randomly divided into six groups for drug administration (details given in Supplementary Table 2). All procedures were carried out in a biosafety cabinet or on a clean bench. We collected data using StudyDirector (version no.: 3.1.399.19; Studylog System, Inc.), including the long and short diameters of each tumor and the bodyweight of each animal. Raw data were directly imported from the digital scale and the vernier calipers. The experiment was terminated when the average tumor volume of the control mice exceeded 2000 mm$^3$ or a week after the last drug administration.

**Statistics and reproducibility**. Statistically significant differences between the mean values were determined using two-tailed Student's *t*-tests. All data are representative of three independent experiments. The values represent the means ± SD and *p*-values < 0.05 were considered statistically significant.

**Reporting summary**. Further information on research design is available in the Nature Research Reporting Summary linked to this article.

## Data availability

Coordinates and structure factor of the structure reported here have been deposited into the Protein Data Bank with PDB Code: 6JJP. The PDB accession codes 4ZQK, 3BP5, 5WT9, 5GGS were also used in this study. Source data for graphs in Figs. 6 and 7 can be found in Supplementary Data 1. The data supporting the findings of the present study are included in the published article and its supplementary information file. All other relevant data are available from the corresponding authors upon reasonable request.

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

## Acknowledgements

We thank the staff of beamline BL17U at Shanghai Synchrotron Radiation Facility (SSRF) for their assistance during data collection.

## Author contributions

J.Z. and M.Z. designed and supervised the study. J.W., R.W., S.J., and S.W. conducted the experiments. M.W. collected the data sets and solved the structures. M.W., J.W., R.W., S.J., S.W., J.Z., and M.Z. analyzed the data and wrote the manuscript.

## Competing interests

The authors declare no competing interests.
