## [Peer Review File · Communications Biology]

Reviewers' comments:

Reviewer #1 (Remarks to the Author):

Wang et al., have developed a new anti-PD1 antibody which blocks PD-L1 and PD-L2 binding. This has become a "hot" area of research given the positive clinical outcome of the "immune checkpoint blockade" therapy approach for the treatment of melanoma, lung, liver, and colon (among many others in active development) tumors. Many anti-PD1, anti-CTLA-4, and anti PD-L1 antibodies have been developed and a few are already approved by FDA and EU regulators for treatment of several tumors. While it is important to develop these tools to prevent ligand-induced activation of negative regulators of T cell activity such as PD1 and CTLA4, it is also questionable how many more do we need, particularly if mechanistically there does not seem to exist many (any?) differences between them. With that in mind, my main concern with the work described in this manuscript is: what is novel about the newly developed antibody?

Antibodies that simultaneously bind protein and carbohydrates are not new, and many have been reported. Mechanistically the experiments reported by the authors are also focused on the "classic" ligand-blockade, so what is left here to describe besides the making of a new antibody?

The experiments reported by Wang et al seem well designed and well executed, the data is reported clearly and while the method section could benefit from having more detail it is generally OK. I have no concerns with the experimental approach taken and the results described.

Major point:

I think the authors may be missing an interesting gateway into PD-1 biology. Put simply, is the glycosylation of Asn68 always the same? if not, how does this affect MW11-h517 binding? Thought experiment, let's imagine that activated CD8s produce a different glycosylation pattern on Asn68 than resting CD8s or CD4 T cells, either because of changes in metabolic processing, changes in protein expression and/or degradation etc. While an investigation of the nature and properties of PD-1 glycosylation is well beyond the scope of this ms, it would be relatively straightforward to perform a very simple experiment that could generate some interesting new biology:

isolate PBMCs, activate them with OKT3 or peptide cocktails and compare PD-1 staining across nivo, pembro, MW11-h317 and non-blocking abs as control, while labeling simultaneously the usual suspects, CD8, CD4, CD3, CD69, CD44, CD25 etc. I think this experiment, or something along these lines, aimed at investigating a bit more about PD-1 biology while taking advantage of the new tool Wang et al has produced (the antibody) could be interesting and provide important insight. Essentially, beyond the pure "blockade", can MW11-h317 unique properties be used to reveal something new about PD-1 and if so, can MW11-h317, even if theoretically, have any sort of advantage over nivo or pembro in future clinical studies?

Minor points:

- 1- what is the origin of the pembro ab?
- 2- the yellow in fig 7 panel c is hard to read, maybe swap for a different tone/color
- 3- according to table S2 the administration duration is 3 weeks, but the plot in fig 7d only goes until 11 days, can the authors explain why?
- 4- it is unclear from the methods, when did the IP injections of the abs started?
- 5- why did the authors chose IP over SC?
- 6- can the authors in the intro or discussion present more examples of antibodies that also recognize carbohydrates and how these have been used?
- 7- the authors mention in p.5 "the affinity of the control nivolumab was 8.06×10^{-9} M. Under the same conditions, MW11-h317 dissociation [k_d (1/s): 9.50×10^{-4}] was greater than nivolumab dissociation [k_d (1/s): 1.69×10^{-3}]"
is this correct? the dissociation rate of MW11-h317 appears slower (smaller k_d), not "greater", correct?
- 8 - what is the affinity of MW11-h317 to PD-1 expressed in bacteria? it would be interesting to know (similar to what has been described in ref 26).

Reviewer #2 (Remarks to the Author):

Wang and colleagues isolated monoclonal antibody MW11-h317 against human PD-1 from mouse hybridomas and solved the crystal structure of the complex of MW11-h317 Fab to human PD-1 extracellular domain at 2.9Å resolution. This manuscript reported MW11-h317 antibody blocks PD-1/PD-L1 and PD-1/PD-L2 interactions. In addition, MW11-h317 has a unique N-linked glycosylation antigen-binding site with antitumor activity. Most parts of the results are convincing and paper overall is nicely written. However, two potent antibodies against PD-1, nivolumab and pembrolizumab, have been approved for cancer therapy by FDA. They need to further expand the utility of their antibody vs. commercially available one, at least in the discussion.

here are some of my comments:

1. In Fig.4, N49A, N74A and N116A mutants of PD-1 showed more binding populations compared to WT PD-1. The variation could be from cell transfection or these three mutants. How to explain? They should examine the affinity of N49A, N74A and N116A mutants with MW11-h317 antibody comparing to WT PD-1 by SPR.
2. In Fig.6, MW11-h317 antibody showed an overlap in PD-1 binding areas compared to nivolumab and pembrolizumab. It is not convincing if nivolumab and pembrolizumab has competition with MW11-h317 antibody. Antibody competition ELISA can be performed to examine if nivolumab and pembrolizumab can completely block MW11-h317 binding to PD-1.

3. In Fig.7, pembrolizumab should be included in all assay. In Fig.7a, it would be great to compare the affinity of PD-L1/L2 vs PD-1 and MW11-h317 vs PD-1. It will show the blocking mechanism of MW11-h317 antibody, higher affinity, space steric or both.

4. In Fig.7d, the end time point of tumor size measurement is Day 11. Tumor size in mice is usually measured until Day20. How about the late time point? The tumor is still not growing in MW11-h317 and nivolumab treatment group? In the previous publications, anti-PD-1 antibody cannot completely inhibit tumor growing in mice. In this study, antibody treatment groups showed completely inhibition. How to explain?

5. Materials and methods do not contain all the details. For example, antibody sequence, cloning vectors, and primers.

Reviewer #3 (Remarks to the Author):

In the present manuscript, Wang et al. present their findings regarding the biophysical properties and the functional features of a PD-1 monoclonal antibody, named MW11-h317. This Ab had high affinity for PD-1 and was effective at blocking PD-1 interaction with PD-L1 or PD-L2 and inducing inhibition of tumor growth in vivo. Moreover, the antibody was capable of effectively inhibiting tumor growth in a mouse model. The crystal structure of a PD-1/MW11-h317 Fab complex showed that both the loops and glycosylation of PD-1 were involved in recognition and binding. The studies identified a unique glycan epitope in PD-1, which allowed for binding to MW11-h317. The unique glycan epitope binding was a feature of the MW11-h317 that was not shared by nivolumab and pembrolizumab. These data suggest that binding of MW11-h317 is critically dependent on the interaction with Asn58 glycan aside from binding by the PD-1 loops. The results are interesting and the development of this antibody might be of significant for the study of PD-1 protein properties and for therapeutic approaches. However, several issues require further attention.

Specific points:

1) It is unclear why the information included in the accompanying manuscript is not part of the manuscript under consideration. The validation of the crystal structure of Ab: PD-1 binding should be part of the present manuscript.

2) The authors stated that their new Ab "specifically targets human PD-1". However, they use it in tumor studies using a mouse model indicating that the antibody also targets mouse PD-1. Appropriate corrections and clarifications are required.

3) Figure 7a: The sensograms of antibody bindings do not seem to have appropriate fitting. Detailed information should be provided.

4) Figure 7b-c: The binding of MW11-h317 and nivolumab equally block the PD1/PD-L1 interaction (Figure 7b). However, the cytokine production data show that MW11-h317 more potently inhibited IFN γ production than Nivolumab. In contrast, IL-2 production was equally inhibited by MW11-h317 and nivolumab (Figure 7c). What is the reason of the selective effect of MW11-h317 on the production of each of these cytokines?

5) Along the same lines, what is the reason of the more potent effect of MW11-h317 and nivolumab on the production of IFN γ when these two antibodies equally blocked the interaction of PD-1 with PD-L1.

6) Figure legend 7: The legend related to panel c is incomprehensive and needs correction.

Dear Editor,

We have studied the comments from referees carefully and tried our best to revise the manuscript. The point-by-point responses to the referees' comments are listed as following:

Reviewer #1 (Remarks to the Author):

Wang et al., have developed a new anti-PD1 antibody which blocks PD-L1 and PD-L2 binding. This has become a "hot" area of research given the positive clinical outcome of the "immune checkpoint blockade" therapy approach for the treatment of melanoma, lung, liver, and colon (among many others in active development) tumors. Many anti-PD1, anti-CTLA-4, and anti PD-L1 antibodies have been developed and a few are already approved by FDA and EU regulators for treatment of several tumors. While it is important to develop these tools to prevent ligand-induced activation of negative regulators of T cell activity such as PD1 and CTLA4, it is also questionable how many more do we need, particularly if mechanistically there does not seem to exist many (any?) differences between them. With that in mind, my main concern with the work described in this manuscript is: what is novel about the newly developed antibody?

Antibodies that simultaneously bind protein and carbohydrates are not new, and many have been reported. Mechanistically the experiments reported by the authors are also focused on the "classic" ligand-blockade, so what is left here to describe besides the making of a new antibody?

Response:

Thank you for the good question. It has been indeed reported about antibodies binding protein and carbohydrates simultaneously. However, glycan modification of PD-1 involved in binding to its targeting antibodies has no report at present. In this study, the binding of MW11-h317 to its target including both the loops and the glycan portions of PD-1, which may increase the specificity and we think it may also reduce the possible off-target effect.

The experiments reported by Wang et al seem well designed and well executed, the data is reported clearly and while the method section could benefit from having more detail it is generally OK. I have no concerns with the experimental approach taken and the results described.

Major point:

I think the authors may be missing an interesting gateway into PD-1 biology. Put simply, is the glycosylation of Asn68 always the same? if not, how does this affect MW11-h517 binding? Thought experiment, lets imagine that activated CD8s produce a different glycosylation pattern on Asn68 than resting CD8s or CD4 T cells, either because of changes in metabolic processing, changes in protein expression and/or degradation etc. While an investigation of the nature and properties of PD-1 glycosylation is well beyond the scope of this ms, it would be relatively straightforward to perform a very simple experiment that could generate some interesting new biology: isolate PBMCs, activate them with OKT3 or peptide cocktails and compare PD-1 staining across nivo, pembro, MW11-h317 and non-blocking abs as control, while labeling simultaneously the usual suspects, CD8, CD4, CD3, CD69, CD44, CD25 etc. I think this experiment, or something along these lines, aimed at investigating a bit more about PD-1 biology while taking advantage of

the new tool Wang et al has produced (the antibody) could be interesting and provide important insight. Essentially, beyond the pure "blockade", can MW11-h317 unique properties be used to reveal something new about PD-1 and if so, can MW11-h317, even if theoretically, have any sort of advantage over nivo or pembro in future clinical studies?

Response:

Thanks a lot for your instructive suggestion. We have only found this one mAb with unique glycan interaction at present which may become a useful tool in PD-1 function research. We will give more attention to it.

Minor points:

1- what is the origin of the pembro ab?

Response:

The origin of pembrolizumab in this study is commercial purchase (Merck &Co., KEYTRUDA® (pembrolizumab), NDC: 0006-3092-02, Lot No.:6SNL81506, Specification: 50mg/Vial).The information was added in Methods section of the revised manuscript.

2- the yellow in fig 7 panel c is hard to read, maybe swap for a different tone/color

Response:

Thanks. It has been corrected in the revised manuscript.

3- according to table S2 the administration duration is 3 weeks, but the plot in fig 7d only goes until 11 days, can the authors explain why?

Response:

Thank you for your careful review. This experiment was accomplished in Crown Bioscience Inc. (Contract Research Organization). The original design of administration duration is 3 weeks. But the tumor of control mice grew too fast in the experiment. According to Animal Welfare, the control mice should be euthanized when the tumor volume exceed 2,000 mm³. So the experiment was terminated early for keeping integrated comparative data of tumor weights. The actual times of administration were 3 and the duration was 11days.

4- it is unclear from the methods, when did the IP injections of the abs started?

Response:

Sorry for that unspecified information in the methods. The IP injection started when the tumor volume was 60-100 mm³. The 0 day in Fig 7d referred to the first IP injection time. We have added this information in methods and figure legends.

5- why did the authors chose IP over SC?

Response:

IP and IV are the common administration routes in mAbs animal tests because of their bio-availability and quick efficacy. We chose IP according to experimental purpose and CRO experiences.

6- can the authors in the intro or discussion present more examples of antibodies that also

recognize carbohydrates and how these have been used?

Response:

Thank you for the good suggestion. We have supplemented the relevant content in the discussion.

Several studies have focused on exploring the effects of glycosylation on the interactions between the antibody and its target, including tumor-specific glycosylation of MUC1, which is important in antigen recognition by MUC1-specific antibodies. Specifically, the SM3 antibody directly binds to the MUC1 glycan epitope, while the high affinity of AR20.5 antibody to MUC1 was due to the glycosylation, which altered the conformational equilibrium of the antigen.

7- the authors mention in p.5 "the affinity of the control nivolumab was 8.06×10^{-9} M. Under the same conditions, MW11-h317 dissociation [kd (1/s): 9.50×10^{-4}] was greater than nivolumab dissociation [kd (1/s): 1.69×10^{-3}]"

is this correct? the dissociation rate of MW11-h317 appears slower (smaller kd), not "greater", correct?

Response:

Thank you for the correction. Our description was not precise. We have corrected in the revised manuscript.

8 - what is the affinity of MW11-h317 to PD-1 expressed in bacteria? it would be interesting to know (similar to what has been described in ref 26).

Response:

Thank you for the question. We did determine the affinity of MW11-h317 to PD-1 expressed in E.coli, which was obviously lower than that produced from mammalian cells. Because the test signals were too weak to fit, the globally fitted curve followed is for reference only. Inconsideration of space limitation, the results were not present in the manuscript.

Sample ID	Loading Sample ID	Conc. (nM)	Response	KD (M)	kon(1/Ms)	kdis(1/s)
Human PD-1 Ecoli	MW11-h317	60	0.0847	4.07E-08	7.37E+04	3.00E-03
Human PD-1 Ecoli	MW11-h317	45	0.0694	4.07E-08	7.37E+04	3.00E-03
Human PD-1 Ecoli	MW11-h317	30	0.0462	4.07E-08	7.37E+04	3.00E-03

Reviewer #2 (Remarks to the Author):

Wang and colleagues isolated monoclonal antibody MW11-h317 against human PD-1 from mouse hybridomas and solved the crystal structure of the complex of MW11-h317 Fab to human PD-1 extracellular domain at 2.9Å resolution. This manuscript reported MW11-h317 antibody blocks PD-1/PD-L1 and PD-1/PD-L2 interactions. In addition, MW11-h317 has a unique N-linked glycosylation antigen-binding site with antitumor activity. Most parts of the results are convincing and paper overall is nicely written. However, two potent antibodies against PD-1, nivolumab and pembrolizumab, have been approved for cancer therapy by FDA. They need to further expand the utility of their antibody vs. commercially available one, at least in the discussion.

Response:

We have expanded the utility of our antibody vs. commercially available antibodies in the discussion as follow:

PD-1 is one of the pivotal molecules of the immune checkpoint therapy. Antibodies targeting PD-1 are the most primary molecules for clinical antibodies combination or bispecific antibodies development in the future. Although nivolumab and pembrolizumab have been approved for cancer therapy by FDA, screening of new therapeutic mAbs targeting PD-1 is beneficial for multiple tumors treatments. MW11-h317 has been proved possessed distinct binding epitopes including not only loops in other PD-1 mAbs but also glucans, which not find in other reported PD-1 mAbs.

here are some of my comments:

1. In Fig.4, N49A, N74A and N116A mutants of PD-1 showed more binding populations compared to WT PD-1. The variation could be from cell transfection or these three mutants. How to explain? They should examine the affinity of N49A, N74A and N116A mutants with MW11-h317 antibody comparing to WT PD-1 by SPR.

Response:

Thanks for your careful review. We also realized this problem and predict it resulted from efficient differences of WT and mutant cell transfection. So we adjusted the results with GFP protein expression to ensure data reliability.

Thank you for the suggestion. We did compare the affinities of mutants with MW11-H317 to WT PD-1 by SPR. The results were consistent with FACS as follows and not present in our manuscript in consideration of space limitation.

MW11-h317					
Sample	Conc.(nM)	Response	KD(M)	kon(1/Ms)	kdis(1/s)
PD-1	60	0.1581	4.92E-09	1.55E+05	7.64E-04
PD-1(N49A)	60	0.1596	6.16E-09	1.44E+05	8.89E-04
PD-1(N58A)	60	0.0285	6.07E-08	2.51E+04	1.52E-03
PD-1(N74A)	60	0.1679	6.45E-09	1.51E+05	9.70E-04
PD-1(N116A)	60	0.1608	8.82E-09	1.19E+05	1.05E-03

2. In Fig.6, MW11-h317 antibody showed an overlap in PD-1 binding areas compared to nivolumab and pembrolizumab. It is not convincing if nivolumab and pembrolizumab has competition with MW11-h317 antibody. Antibody competition ELISA can be performed to examine if nivolumab and pembrolizumab can completely block MW11-h317 binding to PD-1.

Response:

Thank you for the suggestion. We did antibodies competitive ELISA. The results as follows indicated that MW11-h317 was partly competitive with nivolumab or pembrolizumab. We also added the results in Fig. 6 of the revised manuscript.

	IC50(nM)
Pembrolizumab-Fab	24.71
Nivolumab-Fab	29.56
MW11-h317-Fab	18.95

3. In Fig.7, pembrolizumab should be included in all assay. In Fig.7a, it would be great to compare the affinity of PD-L1/L2 vs PD-1 and MW11-h317 vs PD-1. It will show the blocking mechanism of

MW11-h317 antibody, higher affinity, space steric or both.

Response:

Thank you for the suggestion. In our study, we chose one of the commercial PD-1 mAbs nivolumab, which proved PD-1 glycosylation not involved in binding, as control in all assays. In structural comparison, we concerned the binding epitopes differences to both nivolumab and pembrolizumab. We actually analyzed the affinities of PD-1 with PD-L1 and PD-L2. The results as follows are for reference only because these were not the same batch of data and not present in our manuscript.

4. In Fig.7d, the end time point of tumor size measurement is Day 11. Tumor size in mice is usually measured until Day20. How about the late time point? The tumor is still not growing in MW11-h317 and nivolumab treatment group? In the previous publications, anti-PD-1 antibody cannot completely inhibit tumor growing in mice. In this study, antibody treatment groups showed completely inhibition. How to explain?

Response:

Thanks for your careful review.

Q1 & 2: This experiment was accomplished in Crown Bioscience Inc. (Contract Research Organization). The original design of administration duration is 3 weeks. But the tumor of control mice grew too fast in the experiment. According to Animal Welfare, the control mice should be euthanized when the average tumor volume exceed 2,000 mm³. So the experiment was terminated early for keeping integrated comparative data of tumor weights. The actual times of administration were 3 and the duration was 11days.

Q3: The efficacy of PD-1 antibodies is differential due to different model used. The efficacy in PBMC immune reconstituted mice was mild usually, which in most early reports. Now days, transgenic mice are new optional models for pharmacodynamics study. In this study, the mouse model was the commercial PD-1 humanized mouse, which bearing mouse tumor expressing

human PD-L1, such as MC38. The anti-tumor efficacy of PD-1 mAbs was more remarkable in this mouse model.

5. Materials and methods do not contain all the details. For example, antibody sequence, cloning vectors, and primers.

Response:

Thank you for the suggestion. The antibody sequence can be obtained in the PDB under 6JJP. The details of cloning vectors and primers were not shown in consideration of space limitation. The information as follows could be added to the supplementary information if needed.

Table. Primers of antibody h317.

Murine 317 Hchain variable region clone primers	Forward primer	MKV7	ATGGGCWTC AAGATGGAGTCACAKWYYCWGG
	Reverse primer	MCK	TCATCAACACTCATTCTCTgTTgAAgCTCTTgA
Murine 317 L chain variable region clone primers	Forward primer	MHV4	ATGRACTTTGGGYTCAGCTTGRTTT
	Reverse primer	MIgG1CH1	CTAACAAATCCCTgggCACAATTTTCTTgTCCACC
Fused 317 Hchain variable region clone primers	Forward primer	hM-h15	5'-3' CTCAAGCTTAATTGCCGCCACCATG
	Forward primer	SP-317L-F	CCTGGAGCCATCGGAGACATTGTGATGACC
	Reverse primer	SP-317L-R	GGTCATCACAATGTCTCCGATGGCTCCAGG
	Reverse primer	317L-R	GCTGGCGCCGCATCAGCCCGTTTGATTC
Fused 317 L chain variable region clone primers	Forward primer	hM-h15	5'-3' CTCAAGCTTAATTGCCGCCACCATG
		SP-317H-F	GAAGGGCGTGCA GTGCGAAGTGAAGCTGGTG
	Reverse primer	SP-317H-R	CACCAGCTTCACTTCGCACTGCACGCCCTTC

	Reverse primer	317H-R	'GCGCTAGCTGCAGAGACAGTGACCAGAG
--	----------------	--------	-------------------------------

Figure. The vector pKN293E

Reviewer #3 (Remarks to the Author):

In the present manuscript, Wang et al. present their findings regarding the biophysical properties and the functional features of a PD-1 monoclonal antibody, named MW11-h317. This Ab had high affinity for PD-1 and was effective at blocking PD-1 interaction with PD-L1 or PD-L2 and inducing inhibition of tumor growth in vivo. Moreover, the antibody was capable of effectively inhibiting tumor growth in a mouse model. The crystal structure of a PD-1/MW11-h317 Fab complex showed that both the loops and glycosylation of PD-1 were involved in recognition and binding. The studies identified a unique glycan epitope in PD-1, which allowed for binding to MW11-h317. The unique glycan epitope binding was a feature of the MW11-h317 that was not shared by nivolumab and pembrolizumab. These data suggest that binding of MW11-h317 is critically dependent on the interaction with Asn58 glycan aside from binding by the PD-1 loops. The results are interesting and the development of this antibody might be of significant for the study of PD-1 protein properties and for therapeutic approaches. However, several issues require further attention.

Specific points:

1) It is unclear why the information included in the accompanying manuscript is not part of the manuscript under consideration. The validation of the crystal structure of Ab: PD-1 binding should be part of the present manuscript.

Response:

Thank you for the suggestion. According to the suggestion, we provided key validation information in the "Crystallization, data collection, and structure determination" section of "Methods". As the validation report provided by PDB will not be included in the published version, we did not mention this report in the manuscript.

2) The authors stated that their new Ab "specifically targets human PD-1". However, they use it in tumor studies using a mouse model indicating that the antibody also targets mouse PD-1.

Appropriate corrections and clarifications are required.

Response:

Thanks for your review. The mouse model used in tumor studies is PD-1 humanized mice. We inserted the human PD-1 protein coding region into the ATG position of the mouse model, in order to express human PD-1 instead of mouse PD-1 as indicated in methods of manuscript. So there was no mouse PD-1 in the mouse model of our experiments.

3) Figure 7a: The sensograms of antibody bindings do not seem to have appropriate fitting.

Detailed information should be provided.

Response:

Thank you for your suggestion. The Octet QKe system (Fortebio) was used to quantify the binding kinetics of the two antibodies. Langmuir binding model was fitted using Data Analysis 9.0 system-provided. The reasons that the fitting curves seemed not perfect enough may due to time phase differences and the system sensitivity. We did the affinities determination and comparison of MW11-h317 and nivolumab to PD-1 using Biacore. The results as follows and the related contents were replaced in the revised manuscript.

Figure. Affinity comparison of MW11-h317 to nivolumab with PD-1. (a and b). Real time response curves of MW11-h317 (a) and nivolumab (b). Antibody concentrations were 200, 100, 50, 25, 12.5, 6.25 and 3.13 nM respectively. **(c).** Kinetic constants of MW11-h317 and nivolumab interacting with recombinant human PD-1 extracellular region. The affinity of MW11-h317 and nivolumab was determined via SPR on a BiacoreS200 system. Human IgG capture antibody in the standard IgG capture antibody kit was immobilized on a CM5 chip using standard amino coupling kit. Antibody was captured at a certain level (200 Ru here) and reacted with recombinant PD-1 at gradient concentrations (starting with 200 nM sequentially diluted to 3.13 nM) in fluid HBSEP buffer (PH 7.4). At the end of each circle, the captured antibody, along with PD-1, was washed away with regeneration buffer and the chip was used for the next circle reaction until the test was completed. Then, the affinity was calculated in a 1:1(Langmuir) binding fit model by BIAevaluation Software.

4) Figure 7b-c: The binding of MW11-h317 and nivolumab equally block the PD1/PD-L1 interaction (Figure 7b). However, the cytokine production data show that MW11-h317 more potently inhibited IFN γ production than Nivolumab. In contrast, IL-2 production was equally inhibited by MW11-h317 and nivolumab (Figure 7c). What is the reason of the selective effect of MW11-h317 on the production of each of these cytokines?

Response:

Thank you for the question. In our test, IFN γ and IL-2 are the measurement indicators of T cell activity induced by mAbs. The increased secretion of the cell-killing cytokines IL-2 and IFN γ with increased mAbs concentration indicated the activated level of T cell simulated by mAbs. Generally,

the results observed in cellular experiments are the short-term effects of antibody drugs.

5) Along the same lines, what is the reason of the more potent effect of MW11-h317 and nivolumab on the production of IFN γ when these two antibodies equally blocked the interaction of PD-1 with PD-L1.

Response:

Thank you for the question. In this study, MW11-h317 and nivolumab blocked the interaction of PD-1 with PD-L1 similarly while nivolumab had more potent effect on the production of IFN γ . We think that PD-1/PD-L1 blocking activities, cytokines production and in vivo antitumor effect reflect mAbs activities at the molecular, cellular and tissue levels respectively. The factors involved range from simple to complex and the results maybe not consistent. We may pay more attention to in vivo antitumor effect from the point of druggability.

6) Figure legend 7: The legend related to panel c is incomprehensive and needs correction.

Response:

Thank you for the suggestion. We have corrected figure legend 7c in the revised manuscript.

Reviewers' comments:

Reviewer #2 (Remarks to the Author):

thanks for addressing the comments. No further comments.

Reviewer #3 (Remarks to the Author):

In points #4 and #5 of the first round of review, I asked why MW11-h317 and nivolumab that have equal binding affinities and blocking properties of PD-1: PD-L1 interaction (Figure 7b) have distinct capacities to alter cytokine production (Figure 7c). Moreover, there was a discrepancy in the ability of MW11-h317 and nivolumab to revert production of IL-2 vs. IFN γ . Not only the authors' answer is inaccurate because they stated that IL-2 and IFN γ are "cell killing cytokines", but also changed their original data. Original Figure 4c showed that MW11-h317 more potently reverted the inhibition of IFN γ production than nivolumab but MW11-h317 and nivolumab equally reverted IL-2 production. In the revised figure, the data have been changed to show that MW11-h317 reverted both IL-2 production and IFN γ production more potently than nivolumab. No answer is provided why these antibodies that have equal binding affinities and blocking properties of PD-1: PD-L1 interaction (Figure 7b) have distinct capacities to alter cytokine production.

These issues should be properly addressed and the authors should clarify why they changed their original data without providing a relevant notification in their response.

Dear Editor,

We have studied the second round of comments from Reviewer #3 carefully. We are very sorry for the copy/paste mistake during the revision of Fig.7c. The responses are listed as follows:

Dear reviewer,

Thank you very much for the important comments. We must make a sincere apology for the mistake in revised Fig.7c. In the first revision, responding to another reviewer's advice "the yellow in fig 7 panel c is hard to read, maybe swap for a different tone/color", we changed the color/line model in Fig.7c. In the figure recombination, we made copy/paste mistake in figure of IL-2 production with IFN γ production, but we didn't changed any original value. We have already corrected this mistake of Fig.7c in the second revised manuscript.

We also feel sorry for the inaccurate description "cell killing cytokines" about IL-2 and IFN γ , which may be more appropriately described as the indicator of T cell activation and we have corrected in the second revised manuscript.

As to why MW11-h317 and nivolumab have equal binding affinities and blocking properties of PD-1: PD-L1 interaction (Figure 7b) have distinct capacities to alter cytokine production (Figure 7c), We think:

The mechanisms of PD-1 antibodies have been extensively studied during the past decade not only from the fundamental researches but also from the clinical trials. The data of figure 7c were used to prove MW11-h317 could induce a Th1-type cell-mediated immune response. We think that PD-1/PD-L1 blocking activities (Figure 7b), cytokines production (Figure 7c) and *in vivo* antitumor effects (Figure 7d) reflect mAbs biological potency from the molecular, cellular and animal levels respectively. For these studies covering both *In vitro* and *in vivo*, the binding characteristics may play different roles which makes three levels' results were not completely consistent. We speculate that among the molecules with close affinity, better binding speed (higher *kon*) is more advantageous *in vitro* experiments, while better binding stability (smaller *kd*) shows better potency in *in vivo* experiments. In consideration of druggability, we may pay more attention to *in vivo* antitumor effect in current study. We may try to figure out the details about this question in the subsequent research.

Finally, statistical analyses comparing the effects of these antibodies to the effects of isotype control were as follows:

	Conc.	P value	
		MW11-h317	Nivolumab
IFN-γ	2000ng/ml	0.00458	<0.000001
	200ng/ml	0.00053	0.000037
	20ng/ml	0.022688	0.000583
IL-2	2000ng/ml	<0.000001	<0.000001
	200ng/ml	0.003165	0.000001

p<0.0332	*
p<0.0021	**
p<0.0002	***
p<0.0001	****

REVIEWERS' COMMENTS:

Reviewer #1 (Remarks to the Author):

Thank you for the newly submitting manuscript and for addressing some of the previous comments.

The ms is still a bit vague at some points and does not fully substantiate some of the claims that are made. For instance:

- line 121: " Asn58 glycosylation is crucial for MW11-h317 binding to PD-1. " the authors support this by showing that a N58A mutation abrogated antibody binding. While I agree that N58A eliminates the glycosylation, this is not the only thing that changes in a N58A mutation. To support this claim the authors should change the glycosylation but maintain Asn58, which can easily be done by expressing PD-1 in E.coli or using GNTI- mammalian cells.

- line 203: "suggesting that MW11-h317 had a high affinity
204 with human PD-1 protein (Fig. 7a). " this is confusing as in line 201 the actual affinity, KD, is reported. The authors should further add that the Kon appears to be faster for Nivo than MW11-h317 and this is why despite showing better koff, MW11-h317 has a similar KD to Nivo.

- line 214 and 215 "This suggested that MW11-h317 might induce a
215 Th1-type cell-mediated immune response. ", unclear why and how the potential Th1 potentiation effect can be deduced from the in vitro MLR activation used.

- line 332 circle  cycle?

- For SPR, regen buffer conditions should be provided

- paragraph starting at line 358 needs some re-wording, it is not entirely clear what competitive phase and detective phase mean.

- line 389 and 390, were DCs digested or detached from the plate with EDTA?

- lines 601 needs re-wording

- Fig. 6e, Concentrol

- Fig. 7c, concentrtrtion

Reviewer #3 (Remarks to the Author):

The authors promptly responded to my additional questions and provided appropriate clarifications and answers. I think the manuscript is suitable for publication in its current form.

Reviewer #1 (Remarks to the Author):

Thank you for the newly submitting manuscript and for addressing some of the previous comments.

The ms is still a bit vague at some points and does not fully substantiate some of the claims that are made. For instance:

- line 121: " Asn58 glycosylation is crucial for MW11-h317 binding to PD-1. " the authors support this by showing that a N58A mutation abrogated antibody binding. While I agree that N58A eliminates the glycosylation, this is not the only thing that changes in a N58A mutation. To support this claim the authors should change the glycosylation but maintain Asn58, which can easily be done by expressing PD-1 in E.coli or using GNTI- mammalian cells.

Thanks for your professional suggestion.

We have preliminary data addressing the point, but we have acknowledged in the text that the N58A mutation can have other effects as follows:

Although we cannot exclude that the mutation of glycosylation sites have effects other than blocking glycosylation, our data strongly suggest that MW11-h317 is a PD-1 glycosylation-dependent antibody, which is entirely different from previously reported therapeutic antibodies.

- line 203: "suggesting that MW11-h317 had a high affinity

204 with human PD-1 protein (Fig. 7a). " this is confusing as in line 201 the actual affinity, KD, is reported. The authors should further add that the Kon appears to be faster for Nivo than MW11-h317 and this is why despite showing better koff, MW11-h317 has a similar KD to Nivo.

Thanks. We have revised as follows:

MW11-h317 specifically bound to human PD-1 protein with an affinity (K_D) of 3.55×10^{-9} M; the affinity of the control nivolumab was 3.65×10^{-9} M. Under the same conditions, MW11-h317 dissociation [k_d (s^{-1}): 8.43×10^{-4}] was slower than nivolumab dissociation [k_d (s^{-1}): 1.90×10^{-3}], while nivolumab association [k_a ($M^{-1}s^{-1}$): 5.21×10^5] was faster than MW11-h317 [k_a ($M^{-1}s^{-1}$): 2.38×10^5], which make MW11-h317 has a similar K_D to nivolumab. Those results suggested that MW11-h317 had a high affinity with human PD-1 protein (Fig. 7a).

- line 214 and 215 "This suggested that MW11-h317 might induce a

215 Th1-type cell-mediated immune response. ", unclear why and how the potential Th1 potentiation effect can be deduced from the in vitro MLR activation used.

Thanks. We have revised as follows:

Our in vitro experiments by stimulating human mixed lymphocytes indicated that the MW11-h317 increased the secretion of cytokines IL-2 and IFN- γ , the indicators of T cell activation (Fig. 7c), which be considered as classical features of Th1-type cell-mediated immune response. This suggested that MW11-h317 might induce a Th1-type cell-mediated immune response.

- line 332 circle  cycle?

Thanks. Revised.

- For SPR, regen buffer conditions should be provided

Thanks. The buffer condition has been added.

- paragraph starting at line 358 needs some re-wording, it is not entirely clear what competitive phase and detective phase mean.

Thanks. We have revised as follows:

Antibodies competitive ELISA was performed in the same way. Fabs of antibodies, as competitive phase, were treble gradient diluted from 100 to 5.58 nM. MW11-h317, as detective phase, maintained at 0.5 nM. Fabs and MW11-h317 were mixed and then were added and incubated in ELISA plates coating with recombinant human PD-1 protein (residues 21-167). Finally HRP (Horseradish Peroxidase) -anti-human Fc secondary antibody was used to detected how much MW11-h317 binding with PD-1 protein on the ELISA plates.

- line 389 and 390, were DCs digested or detached from the plate with EDTA?

Yes, for mild digestion.

- lines 601 needs re-wording

Thanks. The legend of Fig. 6 has been revised.

- Fig. 6e, Concentrol

Thanks. Revised.

- Fig. 7c, concentration

Thanks. Revised.